# Federated Morozov Regularization for Shortcut Learning in Privacy Preserving Learning with Watermarked Image Data

## ABSTRACT

Federated learning is a promising privacy-preserving learning paradigm in which multiple clients can collaboratively learn a model with their image data kept local. For protecting data ownership, personalized watermarks are usually added to the image data by each client. However, the introduced watermarks can lead to a shortcut learning problem, where the learned model performs predictions over-rely on the simple watermark-related features and represents a low accuracy on real-world data. Existing works assume the central server can directly access the predefined shortcut features during the training process. However, these may fail in the federated learning setting as the shortcut features of the heterogeneous watermarked data are difficult to obtain.

In this paper, we propose a federated Morozov regularization technique, where the regularization parameter can be adaptively determined based on the watermark knowledge of all the clients in a privacy-preserving way, to eliminate the shortcut learning problem caused by the watermarked data. Specifically, federated Morozov regularization firstly performs lightweight local watermark mask estimation in each client to obtain the locations and intensities knowledge of local watermarks. Then, it aggregates the estimated local watermark masks to generate the global watermark knowledge with a weighted averaging. Finally, federated Morozov regularization determines the regularization parameter for each client by combining the local and global watermark knowledge. With the regularization parameter determined, the model is trained as normal federated learning. We implement and evaluate federated Morozov regularization based on a real-world deployment of federated learning on 40 Jetson devices with real-world datasets. The results show that federated Morozov regularization improves model accuracy by 11.22% compared to existing baselines.

## CCS CONCEPTS

• **Computing methodologies** → **Distributed computing methodologies**.

## KEYWORDS

Federated Learning, Watermark, Shortcut Learning

*ACM MM, 2024, Melbourne, Australia*
© 2024 Copyright held by the owner/author(s). Publication rights licensed to ACM.
ACM ISBN 978-x-xxxx-xxxx-x/YY/MM
https://doi.org/10.1145/nnnnnnn.nnnnnnn

## 1 INTRODUCTION

With the growth of applying advanced multimedia technology to commercial applications, concerns about user data privacy have greatly increased [21], and research on privacy-preserving learning has come into being. Federated learning [12, 50] emerges as a promising privacy-preserving learning paradigm, where multiple clients can collaboratively learn a model without exposing their private data to the central server. Federated learning has been widely adopted in many multimedia applications such as medical image classification [27], anomaly detection in public safety surveillance [58], and sentiment analysis in social media content [56].

For data ownership identification and copyright protection, digital watermarking technologies are developed and applied in many multimedia applications [13, 51], through adding the well-designed digital watermark into the image data by the data owner [5, 20]. Training models with the watermarked data may lead to the shortcut learning problem, that is the learned model makes predictions based on the simple shortcut features in the training data, rather than learning the underlying complex core features of the target domain, and presents a good performance on the training dataset but decreased model accuracy on the unseen data [4, 28, 53]. For example, in medical image classification, a trained model detects pneumonia in chest X-rays (CXRs) relying on watermarks that represent which hospital the patient was seen instead of lung pathophysiology used by a radiologist [8, 53].

There are many works proposed to overcome the shortcut learning problem. According to where the shortcut feature is processed, existing works can be divided into data preprocesing-based [30, 34, 36, 42] and regularization-based [18, 32] methods. The data preprocessing-based methods assume the shortcut features of data are useless, and they eliminate the shortcut learning problem by detecting and removing the shortcut features from the training dataset. These methods may fail in learning with the watermarked data as the shortcut features (i.e., the watermark-related features) are important for data ownership identification, and cannot be directly removed in practice. For regularization-based methods, shortcut features are regularized based on certain prior knowledge during each training iteration. For example, FD [18] assumes the shortcut features are represented in specific frequency, and designs a feature-level regularization technique where a randomized filtering layer is applied after each convolution layer to prevent CNNs from learning frequency-specific imaging features. wMMD-T [32] assumes the causal Directed Acyclic Graph (DAG) indicating the relationship between the input image and output label is known and designs a regularizer that leverages knowledge of the causal DAG to efficiently learn a classifier. These regularization-based methods can work well in the centralized learning setting where the central server can directly obtain certain characteristics of the shortcut features. However, they may fail in a federated learning setting with watermarked data, as the characteristics of the watermark

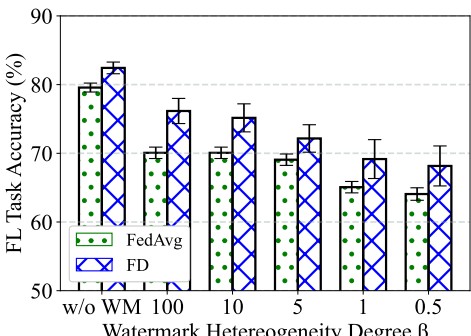

**Figure 1: The task accuracy of a model learned with various watermark heterogeneity.**

features of each client are uncertain and unknown to the server for privacy protection.

Moreover, different clients may apply various digital watermarking techniques on the local data resulting in *watermark heterogeneity*, which further degrades the accuracy of the learned model. Our initial experiments show the impact of watermark heterogeneity under different regularization-based methods. As shown in Fig. 1, with the watermark heterogeneity degree $\beta$ (detail setting can be seen in Sec. 4) increasing from 100 to 0.05, the accuracy of the learned model decreases up to 15.5% under all baselines.

In this paper, we propose a federated Morozov regularization method to solve the shortcut learning problem of learning with watermarked data in a privacy-preserving way. Specifically, we first perform the local watermark mask estimation with the maximum a posteriori (MAP) method to generate the watermark mask, a matrix that can represent the characteristics of the watermarks. We observe that the embedded watermarks with various digital watermarking technologies can all be presented by the *location* and *intensity* map. Therefore, we estimate the watermark mask based on the distinct statistical distributions of natural images and artificial watermarks, capturing the divergence in their spatial and frequency domain characteristics. Then, we aggregate the estimated local watermark mask in the server to generate the global watermark mask with a weighted averaging model. Finally, we perform Morozov regularization-based local training by *actively* adjusting the regularization parameters with the estimated local and global mask. Intuitively, if the model training leads to worse overfitting to shortcut features, the regularization parameter will be increased, i.e., to aggressively mitigate overfitting introduced by the watermark; and vice versa. We evaluate federated Morozov regularization through experiments in real-world settings by deploying it on a test network of 40 Jetson devices, each with varying computational capabilities. We also evaluate our method on a real-world federated watermarked dataset, COVID-FL [52], where watermark heterogeneity is present. Evaluation results demonstrate the superior performance of our method compared to the baselines. federated Morozov regularization improves the accuracy of the learned model by up to 11.22%. We also conducted an ablation study of federated Morozov regularization to validate the contribution of each component to FL model performance in watermarked datasets.

The contributions of this paper can be summarized as:

- We are the first to formulate the shortcut learning problem arising from watermarked datasets in federated learning and find that watermark heterogeneity can further degrade the learning performance.
- We propose federated Morozov regularization, a new regularization method that can automatically adjust the regularization parameters based on the watermark knowledge of all clients in a privacy-preserving way.
- We evaluate federated Morozov regularization by deploying a real-world testbed of 40 Jetson devices with diverse computational capacities and comparing it to several baselines with real-world datasets. Our evaluations show that federated Morozov regularization outperforms existing baselines, achieving 11.22% higher accuracy.

## 2 BACKGROUND & RELATED WORK

### 2.1 FL for Multimedia Application

The integration of federated learning (FL) with multimedia applications is fundamentally motivated by the need to safeguard privacy [23, 29, 55]. This approach has facilitated the advancement of multimedia applications involving personal data, such as image classification [27], anomaly detection in public safety surveillance [58], and sentiment analysis in social media content [56]. The bulk of current research in this area has been concentrated on tackling data-centric challenges [31, 57], including non-iid data [26, 57], data imbalance [44], and the presence of noise [47].

However, a relatively unexplored issue in this domain is the influence of watermarked data in federated learning. Digital watermarking, a strategy widely adopted in multimedia applications for asserting data ownership [13, 51] and copyright protection [5, 20], has found extensive application in data involving privacy and copyright issues, such as medical images [41], surveillance videos [13], and social media [38]. Despite its primary intent, watermarking unintentionally introduces detectable patterns into the data, precipitating a phenomenon known as shortcut learning.

### 2.2 Shortcut Learning

Shortcut learning refers to a phenomenon where deep learning models, during training, preferentially latch onto simple, detectable features—termed as shortcut features—instead of grappling with the more complex, core features of the data [10, 16]. This inclination can lead to models that perform well on training and in-distribution test data but falter significantly when faced with out-of-distribution inputs. Examples of shortcut learning include models relying on background elements [2] or specific textures for image classification [15], and even the presence of watermarks [4, 8].

Solutions to shortcut learning have primarily focused on data prepossessing [30, 34, 36, 42] and regularization techniques [18, 32]. Data prepossessing often involve the removal of shortcut features [34, 36] or data augmentation [30, 42]. However, in federated learning scenarios, watermarks are added due to a lack of trust in the federated learning applications or to embed ownership directly into the training model, making their removal impractical. The regularization method often views shortcuts as a consequence of

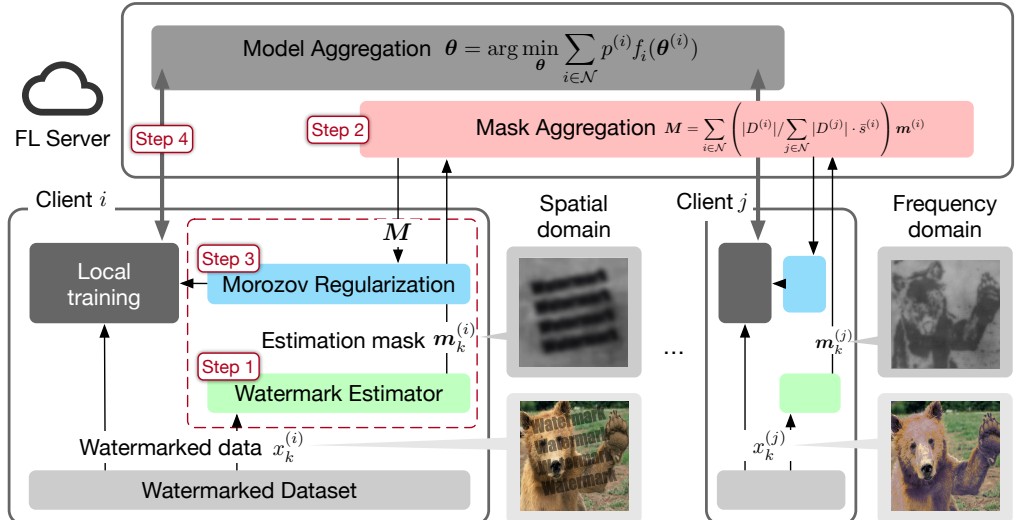

**Figure 2: Overview of the federated Morozov regularization in federated learning.**

model overparameterization [32]. Techniques like FD [18] emphasize high-frequency shortcut features, while methods like wMMD-T [32] focus on background elements as shortcut features. Yet, these shortcut features do not align with those introduced by watermarks.

Moreover, these approaches often require prior knowledge of the shortcut features from client data, such as labels or filter parameters, which contradicts the privacy-preserving nature of federated learning. Utilizing global information from the server side also fails to address the challenges brought by watermark heterogeneity.

## 2.3  Morozov Regularization

Morozov regularization [39] is one type of tool to adjust regularization parameters actively. One key principle of these methods is the discrepancy principle [43]. The rationale is that for a good regularized solution, the norm of the residual should match the noise level of the data.

Morozov regularization has been used in many applications in the past, e.g., to regularize noises from satellite sounder measurements for atmospheric profiling applications [24], to regularize sensor noises in digital images [6] and machine learning [19, 39] recently.

The suitability of Morozov regularization for our problem lies in its precision in targeting specific distributions or explicitly formulated noise, offering localized regularization rather than a blanket, global approach. This characteristic is particularly aligned with the challenges posed by watermarks, which introduce shortcut features localized within parts of an image, rather than affecting it uniformly. Unlike other regularization methods that might operate under broad assumptions about noise or apply regularization uniformly across the entire data set, Morozov regularization provides an adaptive mechanism to fine-tune the regularization parameter, thereby mitigating the shortcut learning effect.

As compared to other regularization, Morozov regularization is simple and has less assumptions on noise approximation [1], practical a-posteriori rules [35], and/or convergence rate [37]. We choose Morozov regularization for its widely applicability and leave other types of regularization into future works.

## 3  FEDERATED MOROZOV REGULARIZATION

### 3.1  Problem Definition

Federated Learning leverages a set of distributed clients $\mathcal{N} = \{1, \ldots, N\}$ to iteratively learn a global model $\boldsymbol{\theta}$ without leaking any private local data to the central server [33]. In each client $i$, the local dataset is defined as $D^{(i)}$. For data ownership identification, each data sample $x_k^{(i)}$ is embedded with a digital watermark $n_k^{(i)}$. Let $\boldsymbol{\theta}^{(i)}$ be the local model of client $i$, and the global model $\boldsymbol{\theta}$ is learned by solving the following optimization problem:

$$F(\boldsymbol{\theta}) := \underset{\boldsymbol{\theta}}{\operatorname{argmin}} \quad \sum_{i=1}^{N} \sum_{k=1}^{D^{(i)}} f^{(i)}(\boldsymbol{\theta}; WM(x_k^{(i)}, n_k^{(i)}), y_k^{(i)}), \quad (1)$$

where $f^{(i)}(\boldsymbol{\theta}^{(i)}) = \frac{1}{|D^{(i)}|} \sum_{(x,y) \in D^{(i)}} \ell(x, y; \boldsymbol{\theta}^{(i)})$, $|D^{(i)}|$ is the number of data sample in client $i$, $WM(\cdot)$ is the watermark embedding function, and $\ell(\cdot)$ is the loss function.

The integration of digital watermarks into image data for ownership identification introduces several challenges in the federated learning environment. Firstly, accurately modeling the embedded watermarks $(n_k^{(i)})$ within the data requires sophisticated techniques to distinguish and quantify their impact on the learning process. Secondly, the distributed nature of federated learning complicates the task of addressing the variability in watermark characteristics across different clients while maintaining global model performance. Thirdly, it is crucial to mitigate the performance degradation caused by watermarks without compromising the privacy of the data.

We propose a federated Morozov regularization method to address these challenges, and Fig. 2 shows the overview of federated Morozov regularization during a training round of the FL model. The complete training process include four steps. After obtaining user consent for training data, the client preprocesses the data online or offline using the *watermark estimator*. This step involves individually inputting watermarked data $x_k^{(i)}$ to obtain the watermark estimation mask $\boldsymbol{m}_k^{(i)}$ for the client's dataset (Step 1). *Mask aggregation* involves aggregating local watermark estimation masks

$m_k^{(i)}$ from each client into a global watermark estimation mask $M$. This step synthesizes collective watermark characteristics from all participating clients (Step 2). Local training using watermarked data is conducted under the *Morozov regularization* module. This module automatically selects regularization parameters based on the watermark estimation mask corresponding to the training data, thereby adjusting the local model parameters to ignore watermarks in the data (Step 3). Finally, the server aggregates the local models from the selected clients to form a new global model for the next round of training (Step 4).

## 3.2 MAP-basd Watermark Mask Estimator

Our stochastic approach is based on maximum a posteriori (MAP) estimation, as described in [48]. Consider the classical problem of watermark embedding, which involves embedding a watermark into an image without considering the image content. In the most general form in communication codec theory [49], the process of $WM(\cdot)$ can be modeled as $x' = x + n$, where $x'$ represents the watermarked data, $x$ is the original data, $x \in \mathcal{R}^N$ with $N = M \times M$, and $n$ denotes the watermark. Our goal is to estimate $\hat{n}$, which is an approximation of the watermark $n$.

Under the general assumption [49], we model the watermark as a Gaussian random variable. Let watermark sample $n_{u,v}(1 \le u, v \le M)$ and image sample $x_{u,v}(1 \le u, v \le M)$ be defined on the vertices of a grid $M \times M$. Further, let all samples be independent and identically distributed, we have conditional probability density of $n_{u,v}$:

$$p_{n_{u,v}}(x_{u,v} \mid n_{u,v}) = \frac{1}{\sqrt{\left(2\pi\sigma_{n_{u,v}}^2\right)^N}} \exp\left\{-\frac{1}{2\sigma_{n_{u,v}}^2}\Delta_n^T\Delta_n\right\}, \quad (2)$$

where the $\sigma_{n_{u,v}}$ of the watermark in the $(u, v)$ location signifies its *intensity*, $\Delta_n = x_{u,v} - n_{u,v}$. Higher variance indicates a more noticeable watermark (albeit with possible image distortion), whereas lower variance results in a subtler watermark.

To estimate the watermark throughout an image, we use a *local estimation mask* $m = [\hat{n}_{u,v}]_{1 \le u,v \le M}$, which is a matrix represent each client's watermark information in local dataset. The index $(u, v)$ in this mask represents the watermark *location*. Each $\hat{n}_{u,v}$ is determined by the MAP criterion:

$$\hat{n}_{u,v} = \text{argmax}_{\tilde{n}_{u,v} \in \mathcal{R}^N}\left(\ln p_{x_{u,v}}(x' \mid \tilde{n}_{u,v}) + \ln p_{n_{u,v}}(\tilde{n}_{u,v})\right), \quad (3)$$

where $\tilde{n}$ represents a hypothetical watermark value being considered during the optimization process to maximize the posterior probability. The estimation accuracy enhancement is due to MAP estimation's statistical convergence towards the true watermark distribution as the dataset grows.

## 3.3 Global Watermark Mask Aggregation

In FL environments, the aggregation of local models is a crucial step for synthesizing a global model that benefits from the distributed learning process. Analogously, the aggregation of local watermark estimation masks is essential for constructing comprehensive global watermark knowledge.

The aggregation of the global watermark estimation mask, denoted by $M$, incorporates contributions from local watermark estimation masks $m^{(i)}$ from each client $i$ within the network $\mathcal{N}$. The aggregation process is governed by the equation:

$$M = \sum_{i \in \mathcal{N}}\left(\frac{|D^{(i)}|}{\sum_{j \in \mathcal{N}}|D^{(j)}|} \cdot \bar{s}^{(i)}\right)m^{(i)}, \quad (4)$$

where the weight for each client's local mask $m^{(i)}$ is determined by the product of two key factors. The first factor, $\frac{|D^{(i)}|}{\sum_{j \in \mathcal{N}}|D^{(j)}|}$, consider the relative data sample size $|D^{(i)}|$ of the $i$-th client, indicating the proportion of data contributed by this client in comparison to the total data volume across all clients in $\mathcal{N}$. The second factor, $\bar{s}^{(i)}$, corresponds to the average size of the watermark estimation mask for the $i$-th client, which is computed as the mean of the dimensions of the mask $m^{(i)}$. This measure reflects the spatial extent of the watermark information present within the client's data.

## 3.4 Morozov Regularization

After obtaining the global watermark estimation mask $M$, it is a *mask integration* with the local masks to refine the watermark knowledge for each client. The refined local mask for client $i$, denoted by $m^{*(i)}$, is achieved by blending $M$ with $m^{(i)}$:

$$m^{*(i)} = \beta^{(i)}M + (1 - \beta^{(i)})m^{(i)}, \quad (5)$$

where $\beta^{(i)} \in [0, 1]$ is an adaptive hyperparameter that controls the degree to which the global mask influences the refined local mask. The value of $\beta^{(i)}$ is dynamically adjusted based solely on the training performance difference, $\Delta\text{Acc}^{(i)}$, which is the difference between the highest validation accuracy among all clients and the validation accuracy of the current client $i$. To ensure $\beta^{(i)}$ scales appropriately between 0 and 1, it is calculated as follows:

$$\beta^{(i)} = \frac{\exp(-\Delta\text{Acc}^{(i)})}{\max_{j \in \mathcal{N}}\exp(-\Delta\text{Acc}^{(j)})}, \quad (6)$$

This formula uses an exponential function to decrease the influence of $\Delta\text{Acc}^{(i)}$ as it increases, ensuring that $\beta^{(i)}$ remains within the desired range and effectively balances the contribution of the global mask based on the relative performance of each client.

Regularization adds a term $reg(\cdot)$ to the loss function $f(\theta; x, y)$, comprising a parameter matrix $\boldsymbol{\alpha}$ and norm $R(\theta)$, formulated as $reg(\theta) = \boldsymbol{\alpha}R(\theta)$. The matrix $\boldsymbol{\alpha}$ balances regularization's importance, with higher values increasing bias and reducing overfitting, and lower values doing the opposite. This balance is captured by $\boldsymbol{\alpha} = \boldsymbol{\alpha}(\delta)$, where $\delta$ measures deviation from real data.

Mathematically, refer to [25], the loss function with regularization is formulated as below,

$$F(\theta^*) := \underset{\theta, \boldsymbol{\alpha}}{\text{argmin}} \sum_{i=1}^{N}\sum_{k=1}^{D^{(i)}}\left(f(\theta; x_k + n_k, y_k) + \alpha||\theta||_2^2\right). \quad (7)$$

Morozov regularization is a principle for choosing a regularization parameter, i.e., $\boldsymbol{\alpha}$, to stabilize the machine learning model to be trained. Specifically, let $x_\alpha$ be

$$x_\alpha = \arg\min_x\left\{\frac{1}{2}||\theta(x) - y||^2 + \boldsymbol{\alpha}R(\theta)\right\}. \quad (8)$$

$\boldsymbol{\alpha}$ can be considered as a control parameter. If $\boldsymbol{\alpha}$ is too small, the model overfits the watermarked in the data; and if $\boldsymbol{\alpha}$ is too big, the model loses the essential details. If $y^{\delta}$ is the watermarked data and assume that $\delta$ is the known noise level introduced by the watermarked data, then $\boldsymbol{\alpha}$ is chosen such that:

$$\|\theta(x_\alpha) - y^\delta\| = \delta = m_k^*. \tag{9}$$

In other words, Morozov regularization chooses the value of $\boldsymbol{\alpha}$ that can make the norm $\| \cdot \|$ equal to the noise level (also called the Morozov's discrepancy principle [43]).

The *federated Morozov regularization* for FL in Alg. 1 operates in three main phases: watermark estimation, mask aggregation, and regularization parameter computation. Initially, each client's model parameters $\theta^{(i)}$ are initialized. The watermark estimation phase involves using MAP-based method to estimate the $\hat{n}_k^{(i)}$ in each data point of client $i$'s dataset $d^{(i)}$ and get the $m^{(i)}$ (Lines 4–7).

The watermark aggregation use the clients' watermark estimation mask to aggregate a global mask with Eq. (4) before federated learning training process (Lines 10–11).

During the federated learning process, each client refines the watermark estimation mask as outlined in Lines 14–17. In this phase, the learning module of each client employs Morozov regularization to compute the regularization parameters. This involves setting an initial discrepancy tolerance and $\boldsymbol{\alpha}_k^{(i)}$, which are iteratively refined based on model predictions $\hat{y}_k^{(i)}$, residuals, and discrepancy measures until they converge within the set tolerance (Lines 18–24).

Finally, the algorithm utilizes the refined $\boldsymbol{\alpha}_k^{(i)}$ to adjust each client's model parameters $\theta^{(i)}$. This adjustment takes into account the loss function $f(\cdot)$, the regularization parameter $\boldsymbol{\alpha}_k^{(i)}$, and the estimated watermark $\hat{n}_k^{(i)}$. Consequently, the regularized loss function $F_{reg}(\theta^{(i)})$ is updated to reflect these changes, ensuring that the model parameters are optimized in alignment with the FL objectives and constraints.

Subsequently, each client performs local model training and adheres to the FL training protocol depicted in Step 4 of Fig. 2. Throughout the FL cycles, Alg. 1 systematically incorporates these updates into the overall FL training scheme.

## 4 EVALUATION

### 4.1 Evaluation Settings

We assess the performance of our technique in a client-server testbed that we have constructed. The server is equipped with an Nvidia RTX 3090 GPU and an AMD Ryzen 9 5900X CPU, running on Ubuntu 20.04 LTS. For client devices, we employ $2 \times$ NVIDIA Jetson AGX Orin, $3 \times$ Jetson Orin Nano 8GB, $5 \times$ Jetson Orin Nano 4GB, and $30 \times$ Jetson Nano. The performance detail can be seen in Table. 1. Our testbed setting and equation[1] have been shown in Fig. 3 to understand the efficacy of federated Morozov regularization in heterogeneity edge clients. We connected each client device to switches via an Ethernet cable. Data exchange in federated learning, including metadata and models, is facilitated by accessing the

---

[1]The 4GB and 8GB Jetson Orin Nano boards have the same appearance.

---

**Algorithm 1:** Federated Morozov Regularization

**Data:** Loss function $f(\cdot)$ in client $i$, data $x^i$ in client $i$.
**Result:** Regularized loss function.

1 **Initialization:**
2 **for** *client i in $\mathcal{N}$* **do**
3     **for** *data $x_k^{(i)}$ in $D^{(i)}$* **do**
4         Compute $m_k^{(i)}$ using Eq.(3);
5     **end**
6     Combine dataset mask $m^{(i)} = \frac{1}{|D^{(i)}|} \sum_{k=1}^{|D^{(i)}|} m_k^{(i)}$;
7     Upload $m^{(i)}$ to server;
8 **end**
9 **Server:**
10     Aggregate $M$ using Eq. (4) and $\{m^{(i)}\}_{i=1}^{\mathcal{N}}$ from clients;
11     Broadcast global watermark mask $M$ to each client;
12 **Start FL training**:
13 **for** *each client i in $\mathcal{N}$* **do**
14     Upload last round's local Acc.;
15     Calculate $\Delta Acc^{(i)}$ before received global Acc. from server ;
16     Update $\beta^{(i)}$ using Eq. (6);
17     Refine $m^{*(i)}$ using Eq. (5);
18     Apply watermark estimation $\hat{n}_k^{(i)} \leftarrow m_k^{*(i)}$;
19     Initialize value for $\boldsymbol{\alpha}_k^{(i)}$ and set tolerance *tol*;
20     **while** *discrepancy > tol* **do**
21         Compute the model prediction $\hat{y}_k^{(i)} \leftarrow f(\theta^{(i)}; x_k^{(i)})$;
22         Compute the residual: $residual_k^{(i)} = \left\| y_k^{(i)} - \hat{y}_k^{(i)} \right\|_2^2$;
23         Compute the discrepancy:
        $discrepancy = residual_k^{(i)} - \left\| \hat{n}_k^{(i)} \right\|_2^2$;
24         Update $\boldsymbol{\alpha}_k^{(i)}$;
25     **end**
26     **return** $\boldsymbol{\alpha}_k^{(i)}$;
27     **Update Regularized Loss:**
    $F_{reg}(\theta^{(i)}) \leftarrow f(\theta^{(i)}; x_k^{(i)} + \hat{n}_k^{(i)}, y_k^{(i)}) + \boldsymbol{\alpha}_k^{(i)} \|\theta^{(i)}\|_2^2$;
28 **end**

---

IP bound to each device. The communications protocol uses sockets. The underlying Jetson driver is supported by Jetpack 5.1.

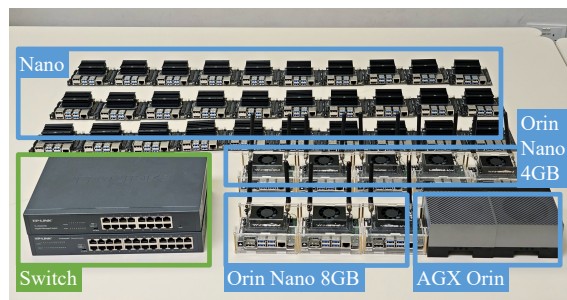

**Figure 3: Evaluation testbed in the lab.**

**Table 1: Performance Comparison of Edge Devices**

| Device | Quantity | GPU | CPU |
|---|---|---|---|
| AGX Orin | 2 | 248 TOPS | 8-core, 2.2 GHz |
| Orin Nano-8 | 3 | 40 TOPs | 6-core, 1.5 GHz |
| Orin Nano-4 | 5 | 20 TOPS | 6-core, 1.5 GHz |
| Nano | 30 | 472 GFLOPS | 4-core, 1.4 GHz |

**Models and datasets.** In the experimental phase, our investigation employed a range of neural network architectures to perform the image recognition task. The first is a Lite-CNN[2], characterized by its simplicity yet effectiveness. Alongside our custom CNN, we integrated two well-established models: VGG [45] and ResNet-18 [14]. The initial learning rate is set to 0.1, and the batch size is set to 64 by default. Using the SSP [17] synchronization strategy, the local epochs are set to 5 by default. In our experimental setup, we evaluated the performance of our proposed method using widely recognized image classification datasets, including MNIST [9], Cifar-10 [22], Tiny-ImageNet [7] and COVID-FL [52]. In our experimental design, MNIST was trained using the Lite-CNN to assess the effectiveness of our method on simple tasks with a straightforward model. For Cifar-10, we employed the VGG model to train, leveraging its depth and complexity for a more detailed image classification task. The Tiny-ImageNet, serving as a multi-classification challenge, and the binary classification task of COVID-FL were both approached using the ResNet architecture, to handle the complexity and scale of these datasets adequately.

**Watermark setting.** In the creation of our watermarked dataset, various watermark embedding techniques, including frequency watermark: DWT, DCT, DFT, LSB, and spatial watermark: LSB, explicit watermarking [3, 41], are employed. Although the adjustment parameters for watermark intensity vary across different methods (for instance, the intensity in explicit watermarks refers to transparency, while in some frequency domain watermarks, like DFT, it refers to modulation amplitude), we normalize the intensity of all watermarks to a 0-1 scale. The embedding location in spatial domain watermarks denotes the position of the watermark within the image (such as the center or edges), whereas in frequency domain watermarks, it refers to the frequency within the image spectrum (like high, mid, or low frequency; in DCT, this ranges from the LL to HH domain).

The watermark embedding intensity is adjusted between 0.01 and 1 for our experiments. Each client in the FL employs the same watermarking method and parameters, ensuring consistency across the dataset.

**Evaluation metrics.** Our performance evaluation focuses FL performance. Referring to the evaluation metric of the shortcut learning research [18, 32, 36], FL performance is assessed using task accuracy, which measures the percentage of correct predictions by the FL models on a distributed dataset, and loss, indicating the prediction error with lower values signifying better performance.

**Benchmark methods.** We compare the federated Morozov regularization with the following peer robust training methods in

---

[2]Lite-CNN comprises two convolutional layers, each with a 5x5 kernel size and 64 channels, succeeded by a 3x3 max pooling layer. The network also includes two fully connected dense layers, the first containing 384 units and the second 192 units, culminating in a softmax output layer for classification.

---

FL [40] [11], generalized regularization [54] and regularization for shortcut learning [18].

- **FedAvg [33]:** Used to establish a performance baseline in our experiments, serving as a foundation for comparison with other FL algorithms.
- **GroupLasso [54]:** a generalized regularization for machine learning by adding a penalty term. We modified GroupLasso to federated learning version based on the client-level profiling setting.
- **AFL [11]:** Using global model transmission, local gradient calculations, and averaging, with hyperparameters set to $\alpha_1 = 0.75, \alpha_2 = 0.01, \alpha_3 = 0.1$ in our experiments.
- **RFA [40]:** A Roubstness aggregation method for corrupted data. We applied with hyperparameters as per the original paper: R = 3 and $v = 10^{-6}$.
- **FD [18]:** A feature regularization with frequency filter tools. We modified FD to federated learning version (Fed-FD) based on the client-level profiling setting.

## 4.2 Evaluation Results & Analysis

*4.2.1 Improvement with federated Morozov regularization.* The evaluation metric was task accuracy in FL, compared under two different training and inference conditions: with watermarked data but clean inference, and with both watermarked training and inference.

The experimental design, as outlined in Table 2, bifurcates the analysis into two primary scenarios: inference on clean data and inference under watermarked conditions. This distinction aims to uncover the impact of shortcut learning induced by watermarks, which affects not only the inference with watermarked features but also the performance on clean data, highlighting the pervasive influence of watermarks on model behavior. The settings for data and watermark heterogeneity are set to $\alpha = 0.5, \beta = 0.5$, which be defined in Sec. 4.2.2.

Our method demonstrates superior accuracy across all datasets and settings, underscoring its effectiveness in mitigating the adverse effects of shortcut learning in FL. Specifically, in the clean inference setting, our approach achieves an accuracy of 97.35% on MNIST, 80.86% on Cifar-10, 35.43% on ImageNet, and 87.14% on COVID-FL. These results are notably higher than those obtained with other methods, such as *FedAvg, GroupLasso, AFL, RFA,* and *Fed-FD*. The improvement is even more pronounced in the watermarked dataset & inference setting, with scores of 95.24% on MNIST, 79.26% on Cifar-10, 33.29% on ImageNet, and 84.10% on COVID-FL. The detail learning performance with epoch growing can be seen in Fig. 4.

The underperformance of other methods can be attributed to their inability to effectively address the dual challenge posed by non-IID data and the presence of watermarks. Methods like *FedAvg* and *GroupLasso*, while foundational in FL, lack specific mechanisms to counteract the nuanced effects of watermarked data, leading to compromised accuracy. *AFL* and *RFA*, despite introducing robustness in aggregation, do not directly tackle the issue of shortcut learning induced by watermarks. *Fed-FD*, which applies feature regularization, shows promise but still falls short of fully mitigating the impact of watermarks on model learning.

*4.2.2 Results on Data and Watermark Heterogeneity.* In FL, the issue of data-level heterogeneity is primarily manifested through

**Table 2: FL method benchmark accuracy(%) comparison under different settings.**

| Setting | Watermarked Dataset & Clean Inference | | | | Watermarked Dataset & Inference | | | |
|---|---|---|---|---|---|---|---|---|
| | MNIST | Cifar-10 | ImageNet | COVID-FL | MNIST | Cifar-10 | ImageNet | COVID-FL |
| FedAvg | 95.35±0.04 | 71.29±0.72 | 25.56±1.06 | 77.30±1.42 | 92.10±0.02 | 64.07±0.83 | 15.45±1.14 | 74.43±1.23 |
| GroupLasso | 96.44±0.02 | 71.30±0.71 | 28.94±1.00 | 81.43±1.53 | 92.59±0.01 | 64.14±0.51 | 19.22±0.83 | 74.29±1.43 |
| AFL | 96.05±0.02 | 72.52±0.35 | 30.42±0.53 | 83.41±2.56 | 91.53±1.03 | 65.10±0.07 | 22.52±0.51 | 75.41±1.51 |
| RFA | 96.73±0.03 | 72.54±0.87 | 30.62±1.03 | 84.52±1.98 | 94.14±0.01 | 67.89±1.12 | 25.70±0.97 | 77.62±1.83 |
| Fed-FD | 96.86±0.01 | 76.89±0.75 | 33.80±0.71 | 84.09±1.27 | 93.83±0.02 | 68.04±0.90 | 24.03±0.95 | 79.93±1.25 |
| **Ours** | **97.35±0.03** | **80.86±1.04** | **35.43±0.73** | **87.14±1.03** | **95.24±0.02** | **79.26±1.09** | **33.29±1.01** | **84.10±1.68** |

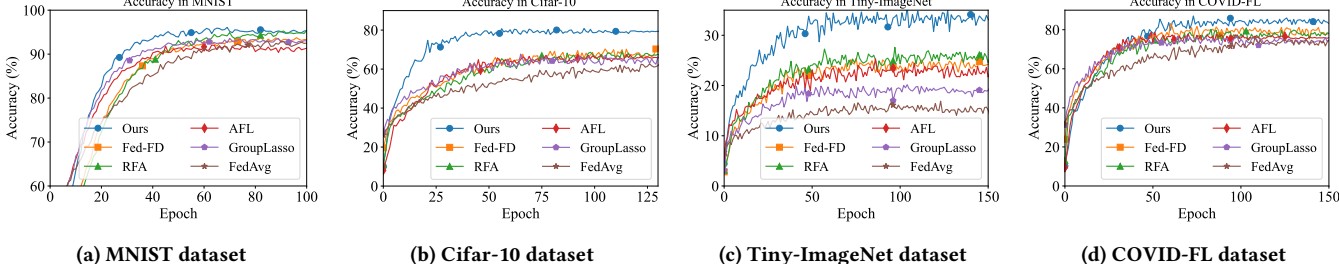

(a) MNIST dataset     (b) Cifar-10 dataset     (c) Tiny-ImageNet dataset     (d) COVID-FL dataset

**Figure 4: Task accuracy and convergence with epoch growing of method benchmark in different watermarked datasets.**

the presence of non-IID (independent and identically distributed) data challenges. For the non-IID problem in the FL experiment, we define the degree of non-IID data and non-IID watermark as follows:

In a multi-client training scenario, each client's data is independently sampled with class labels from $N$ classes, following a categorical distribution with vector $q$ ($q_i \geq 0$, $i \in [1, N]$, $\|q\|_1 = 1$). Non-IID client data is simulated by sampling $q$ from a *Dirichlet distribution*, $\text{Dir}(\alpha \mathbf{p})$, where $\mathbf{p}$ is the prior class distribution, and $\alpha > 0$ determines client similarity. An infinite $\alpha$ implies uniform client distributions, while $\alpha$ near zero results in maximum divergence among clients.

In the context of non-IID watermark settings, we adopt a distribution similar to the *Dirichlet distribution* to manage the variability in watermark characteristics such as intensity ($I$) and location ($L$). Intensity ranges from 0 (no watermark) to 1 (maximum intensity), while location varies from low-frequency areas or image edges to high-frequency areas or central regions. We introduce a parameter $\beta$ in $\text{Dir}(\beta \mathbf{p})$ to control the degree of non-IID in the watermark distribution. A higher $\beta$ indicates more uniformity in watermark characteristics across clients, leading to similar intensity and location settings. Conversely, a lower $\beta$ results in greater diversity, with each client having distinct watermark intensity and placement. This approach allows us to simulate a spectrum of watermark patterns across different clients, reflecting various degrees of intensity and placement. Unlike other datasets, for the real-world dataset COVID-FL, the data is already divided among different clients by medical institutions, thus we utilize the official non-IID configuration distribution to proceed. The variation in equipment used by different medical institutions, along with their respective watermark design preferences, inherently introduces a non-IID distribution of watermarks. Therefore, COVID-FL, as a more realistic watermarked

dataset, can be considered a reference for real-world issues and does not require additional non-IID watermark design and settings.

In our experimental analysis, the combined impact of non-IID data and non-IID watermark on the federated Morozov regularization technique is depicted through heatmaps, revealing a compounded decrease in accuracy with the simultaneous presence of both non-IID conditions. We have selected *Fed-FD* as the benchmark for testing our method based on its superior performance as demonstrated in Sec. 4.2.1. When the non-IID degree for both data and watermark is at its highest, we observe a notable reduction in accuracy, illustrating the challenges posed by these conditions. For example, with $\alpha$ of 0.5 and $\beta$ of 0.5, the accuracy drops to around 68.04%. Modifications to the technique, as reflected in the second heatmap, show improvements in this challenging scenario with a notable increase in accuracy. Under the same high non-IID conditions, the accuracy improves to 79.26%. The third heatmap, which focuses on the percentage of improvement, highlights the effectiveness of our modifications. In scenarios with non-IID data and watermark, our method achieves a substantial improvement, with the most pronounced increase in accuracy reaching up to 11.22%.

*4.2.3 Ablation Study.* We present an analysis of three components designed for such environments: MAP-based watermark estimation (MAP), watermark estimation aggregation (Aggr.) and Morozov regularization (Moro.) in Table. 3. The goal is to evaluate how effectively these components, can counteract the reduction in accuracy often caused by watermarking, compared to alternative methods or variations.

**Study on watermark estimation.** Our exploration delved into the efficacy of MAP-based watermark estimation by comparing it against both its variants and analogous statistical methodologies. This included the Blind Image Quality Measurement (denoted

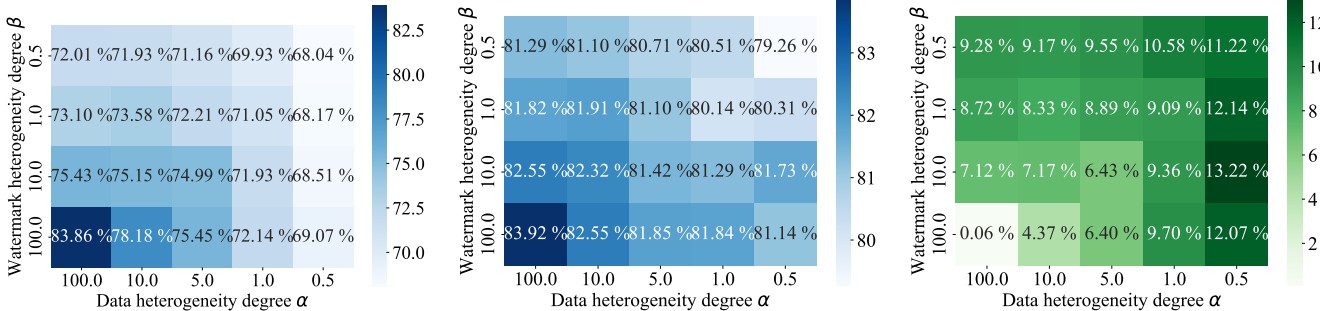

(a) Accuracy with Fed-FD    (b) Accuracy with our method    (c) Improvement between 5a and 5b

**Figure 5: Accuracy in heterogeneous FL environment with Cifar-10 dataset.**

**Table 3: Accuracy(%) comparison in ablation study. The bottom line is the component of our method.**

**(a) Study on watermark estimation.**

| Method | MNIST | Cifar-10 | ImageNet | COVID-FL |
|--------|-------|----------|----------|----------|
| Blind | 93.86 | 64.89 | 26.80 | 79.00 |
| Stacking | 93.55 | 66.58 | 26.29 | 81.26 |
| **MAP** | **95.24** | **79.26** | **33.29** | **84.10** |

**(b) Study on estimation mask aggregation.**

| Method | MNIST | Cifar-10 | ImageNet | COVID-FL |
|--------|-------|----------|----------|----------|
| w/o. | 94.45 | 72.52 | 30.50 | 83.23 |
| Avg. | 95.05 | 73.57 | **33.42** | 83.50 |
| **Aggr.** | **95.24** | **79.26** | 33.29 | **84.10** |

**(c) Study on feature extractor regularization.**

| Method | MNIST | Cifar-10 | ImageNet | COVID-FL |
|--------|-------|----------|----------|----------|
| Tik. | 94.24 | 73.44 | 31.46 | 80.29 |
| L1 | 93.93 | 76.25 | 30.21 | 80.41 |
| **Moro.** | **95.24** | **79.26** | **33.29** | **84.10** |

as Blind) [46], a technique predicated on leveraging statistical attributes to gauge image quality, and the strategy of stacking all dataset images to generate a uniformly weighted mask, tantamount to an averaged weighted MAP-based estimation (denoted as Stacking). As delineated in Table 3a, the MAP approach manifested a notably superior accuracy enhancement relative to its counterparts, with a 12.68% increment over Stacking within the Cifar-10 dataset. Such findings underscore that methodologies centered on image quality estimation (Blind) and indiscriminate estimation of images and watermarks (Stacking) are ineffectual in procuring a robust watermark estimation.

**Study on estimation mask aggregation.** We delve into the efficacy of watermark mask aggregation by both omitting this component (denoted as w/o.) and evaluating its variants, specifically average aggregation (denoted as Avg.), where the local masks from all clients undergo aggregation with equal weighting. As evidenced in the Table. 3b, aggregation demonstrates enhanced performance

in the Cifar-10 dataset, characterized by strong heterogeneity and a smaller quantity of images. Conversely, for datasets with a larger volume and more uniform data, such as Tiny-ImageNet and COVID-FL, the performance difference compared to average aggregation is minimal. This phenomenon can be attributed to the intrinsic purpose of mask aggregation, which is to furnish a global mask that aids clients with less data in obtaining a more applicable mask. Therefore, if the local datasets of clients are sufficiently large, the improvement brought about by aggregation may be marginal.

**Study on Morozov regularization.** In our ablation study focusing on Morozov regularization, we maintained identical inputs for the estimation mask while employing a simplified form of regularization. Morozov regularization, conceptualized as a variant of Tikhonov regularization, introduces parameter adjustments that are more finely tuned to the noise levels encountered. Thus, Tikhonov regularization (denoted as Tik.) is utilized as a comparative measure to ascertain the significance of adjustments in regularization parameters. Furthermore, we investigate whether L1 regularization, a widely referenced regularization technique, also demonstrates improvements in the context of prior information on watermark estimation (denoted as L1). Insights from Table. 3c reveal that the enhancements attributed to Morozov Regularization are predominantly observed in datasets with smaller capacities, such as Cifar-10, and in datasets where the watermark patterns are relatively fixed, such as COVID-FL. It is also observed that, although other forms of regularization exhibit limited improvements over the baseline, their compatibility with watermark estimation is not as pronounced.

## 5 CONCLUSION

Our introduces federated Morozov regularization, a technique tailored for federated learning training on watermarked data. Addressing the challenges posed by the diversity in watermarking algorithms and intensities across FL participant devices, federated Morozov regularization efficiently facilitates FL without requiring prior knowledge of these watermark specifics. The system's ability to probe watermark details and employ Morozov regularization for adapting local model training to watermarked data sets it apart. Our extensive experiments, conducted on a testbed of 40 Jetson edge devices. Federated Morozov regularization improves the accuracy by 11.22%. We also conducted an ablation study of federated Morozov regularization to validate the contribution of each component to FL model performance in watermarked datasets.

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
