# OpenReview forum: "Federated Morozov Regularization for Shortcut Learning in Privacy Preserving Learning with Watermarked Image Data"
_acmmm.org/ACMMM/2024/Conference — MM2024 Poster_

### Official Review · Reviewer_gdM4 · 2024-05-20

**Rating:** 4
**Confidence:** 2

**Summary:**

This paper proposes a federated Morozov regularization technique, where the regularization parameter can be adaptively determined based on the watermark knowledge of all the clients in a privacy-preserving way, to eliminate the shortcut learning problem caused by the watermarked data. Specifically, federated Morozov regularization firstly performs lightweight local water mark mask estimation in each client to obtain the locations and intensities knowledge of local watermarks. Then, it aggregates the estimated local watermark masks to generate the global watermark knowledge with a weighted averaging. Finally, federated Morozov regularization determines the regularization parameter for each client by combining the local and global watermark knowledge. With the regularization parameter determined, the model is trained as normal federated learning.

**Strengths:**

1.	This paper is the first to formulate the shortcut learning problem arising from watermarked datasets in federated learning and find that watermark heterogeneity can further degrade the learning performance.
2.	This paper proposes federated Morozov regularization, a new regularization method that can automatically adjust the regularization parameters based on the watermark knowledge of all clients in a privacy-preserving way.
3.	The authors evaluate federated Morozov regularization by deploying a real-world testbed of 40 Jetson devices with diverse computational capacities and comparing it to several baselines with real-world datasets.

**Limitations:**

1.	Evaluate the impact of varying degrees of Non-i.i.d data and watermarking across different datasets on the experimental results. Assessing only CIFAR10 is not sufficient.

2.	Why use VGG for CIFAR10 and ResNet for Tiny-ImageNet? ResNet also performs well on CIFAR10.

**Suitability:**

3

---

### Official Review · Reviewer_DCUv · 2024-05-22

**Rating:** 5
**Confidence:** 3

**Summary:**

This paper proposes a Federated Morozov Regularization method to address the shortcut problem caused by watermarked data in federated learning scenarios. It models the watermark using Gaussian random variable and estimates the watermark with position and density information based on maximum a posteriori estimation. The server aggregates the local watermark information uploaded by clients to obtain global watermark information. Clients then adaptively generate regularization parameters based on the received global watermark information and local watermark information for model training. The paper effectively eliminates the shortcut problem caused by watermarks in federated learning while preserving privacy, performing well under both non-IID data and watermark conditions.

**Strengths:**

-	Novel approach: The authors address the shortcut problems introduced by watermark data in federated learning scenarios and introduce Morozov Regularization method for diverse watermark techniques to mitigate their impacts on model performance.
-	Theoretical statement: The author defines the problem being addressed and provides a detailed mathematical formulation of the three key steps, facilitating a comprehensive understanding of the design.
-	Adequate evaluation and promising results: The authors conduct extensive experiments using four datasets, considering various watermark embedding techniques and the heterogeneity of both data and watermarks. The evaluation results demonstrate that the proposed method outperforms baselines and performs well under non-IID data and watermark conditions.
-	The paper is well-written and easy to follow.

**Limitations:**

-	Watermark setting: The paper experiments with various watermarking techniques， but appears to only consider diversity in intensity and position of the same watermark. However, in real-world scenarios, participants may utilize different watermarking methods among themselves and internally. The authors should provide reasons for focusing solely on diversity within the same watermarking method and further specify the scope of the paper.
-	Experiments: The experimental design could be expanded to include three primary scenarios, leveraging clean datasets to better elucidate the impact of watermarked data through inference.
-	Writing errors: The "return" statement in line 26 of the algorithm 1 appears unnecessary and may cause subsequent statements not to be executed. The phrase "Our introduces" in the beginning of the conclusion lacks a subject.

**Suitability:**

2

---

### Official Review · Reviewer_2VpS · 2024-06-01

**Rating:** 3
**Confidence:** 3

**Summary:**

The paper discusses a method to improve federated learning, a collaborative model training approach that keeps data local to ensure privacy. In federated learning, clients often add personalized watermarks to their image data to protect data ownership. However, these watermarks can create a shortcut learning problem where the model relies too heavily on the watermark features, reducing its accuracy on real-world data.

To address this issue, the authors propose a federated Morozov regularization technique. This method adaptively determines a regularization parameter based on the watermark information from all clients in a privacy-preserving manner. The process involves:

- Local Watermark Mask Estimation: Each client estimates the locations and intensities of their watermarks.
- Global Watermark Knowledge Aggregation: The estimated local watermark masks are aggregated using weighted averaging to generate global watermark knowledge.
- Regularization Parameter Determination: Combining local and global watermark knowledge, the regularization parameter for each client is determined.

With this parameter, the model is trained as in standard federated learning. The method was implemented and evaluated on a real-world deployment with 40 Jetson devices and real-world datasets. Results show that federated Morozov regularization improves model accuracy by 11.22% compared to existing methods.

**Strengths:**

- The paper provides detailed architectural diagrams, mathematical derivations, and algorithms to illustrate the methodology's process.
- The experimental section in the paper is clear and well-explained, including relevant photos of the experimental machines to ensure the authenticity of the experiments.
- The evaluation section presents data clearly, demonstrating the proposed method's advantages in improving accuracy and training speed.
- The article includes an ablation study, examining the potential impacts of watermark estimation, estimation mask aggregation, and Morozov regularization on the experiments.

**Limitations:**

- This paper focuses on the optimization of federated shortcut learning and watermarked data, with relatively weaker contributions to aspects such as transmission, processing, and presentation of multimedia and unimedia data.
- In the Introduction section, page 2, line 141, the content is "increasing from 100 to 0.05," but the actual is "increasing from 100 to 0.5."
- In the sentence, "Moreover, different clients may apply various digital watermarking techniques on the local data resulting in watermark heterogeneity, which further degrades the accuracy of the learned model," the authors should list the specific digital watermarking techniques used as background knowledge for readers.
- The authors should add the reason for using maximum a posteriori (MAP) estimation for the watermark mask instead of other methods.
- In the dataset selection, the authors should include information about the data volume and characteristics, such as the number of images, image sizes, and label types, and explain the significance of choosing these datasets.
- The authors used different neural network models for different datasets. Although the intent might be to demonstrate the general applicability of federated Morozov regularization, this approach could impact the results when using models of different scales. If possible, an ablation study should be added to address this.
- Issues with Figure 5:
    - The font size of the percentage numbers is too large, causing some overlapping and affecting readability.
    - The figure's location is not indicated in the text.
    - The three subplots do not have a unified color scheme, nor is there an explanation for the color choices.
    - In caption (c): "Improvement between 5a and 5b," the text does not explain what 5a and 5b are.

**Suitability:**

2

---

### Official Review · Reviewer_4CM2 · 2024-06-01

**Rating:** 4
**Confidence:** 2

**Summary:**

This paper proposes a federated Morozov regularization technique to address shortcut learning from watermarked data in federated learning. Tested on 40 Jetson devices, this method surpasses existing baselines by a large margin.

**Strengths:**

1. This paper is well-organized and clear.
2. The motivation is good and the idea is interesting.
3. Experimental results are promising.

**Limitations:**

As I am not very familiar with this field, could you provide a detailed explanation of how this paper relates to multimedia?

**Suitability:**

2

---

### Meta-Review · Area_Chair_7uya · 2024-07-01

**Recommendation:** Accept (Poster)
**Confidence:** 4

**Metareview:**

Summary:

This paper proposes a federated Morozov regularization technique to address shortcut learning from watermarked data in federated learning. The method was tested on 40 Jetson devices and demonstrated significant improvements over existing baselines.

Strengths:
1. The paper is well-organized and clearly presents the proposed method.
2. The motivation for addressing shortcut learning in federated learning using watermarked data is strong, and the idea is interesting.
3. Experimental results show that the proposed method outperforms existing techniques.

Limitations:
1. The paper should provide more details about the digital watermarking techniques used and justify the use of maximum a posteriori (MAP) estimation for watermark masks.
2. The authors should include information about the datasets' volume and characteristics, and justify their choice.
3. The evaluation should consider the impact of non-i.i.d data and watermarking across different datasets, not just CIFAR10. The use of different neural network models for different datasets should be explained.

After the rebuttal, one reviewer raised the final rating, so all reviewers tend to accept this submission. Therefore, I will recommend this paper as Accept (Poster).